# Experimental Study on the Cracking and Mechanical Properties of Lime Soil with Different Slaking Conditions of Newly Repaired Earthen City Walls

**DOI:** 10.3390/ma15124151

**Published:** 2022-06-11

**Authors:** Jianwei Yue, Huicong Su, Xiao Song, Xiangchun Xu, Limin Zhao, Gang Zhao, Peng Li, Ying Chen

**Affiliations:** 1School of Civil Engineering and Architecture, Henan University, Kaifeng 475004, China; yjwchn@126.com (J.Y.); a2104939923@163.com (H.S.); mumu55882005@163.com (L.Z.); chipren@163.com (P.L.); chenying135231@126.com (Y.C.); 2Kaifeng Key Laboratory of Restoration and Safety Evaluation of Immovable Cultural Relics, Kaifeng 475004, China; 3Henan Institute for the Protection of Cultural Relics and Buildings, Zhengzhou 450002, China; zh.hn@126.com

**Keywords:** lime soil, slaking condition, crack control, slaking reaction, particle gradation

## Abstract

In this paper, a method to control the lime reaction by different slaking conditions is proposed to reduce the occurrence of cracks in newly repaired earthen city walls. The effects and mechanisms of the slaking time (0 h, 12 h, 24 h, 48 h and 72 h), lime content (10%, 15% and 20%), and moisture content (14%, 18% and 22%) on the cracking and mechanical properties of lime soil were analyzed by the test results of surface cracks, triaxial compression, particle gradation, pH value, X-ray diffraction and scanning electron microscope. The results show that proper slaking of lime soil specimens can reduce surface cracks and improve mechanical properties. After 12 h of appropriate slaking, the crack rate of the lime soil with 20% content decreased by 97.13%, the cohesion increased by 20.27%, and the internal friction angle decreased by 11.27%. However, the mechanical properties decreased when the slaking time was too long. After 72 h of slaking, the cohesion of 20% lime soil decreased by 8.21% and the internal friction angle increased by 2.82%. Further analysis shows that the appropriate slaking conditions can regulate the reaction rate and alkali environment, control the lime produced cementitious substances, improve the particle gradation and further reduce the occurrence of surface cracks. These results provide a basis for the restoration technology of newly repaired earthen city walls.

## 1. Introduction

Lime and soil are mixed to form lime soil, which has the characteristics of high strength, good integrity, economic material and convenient construction [1,2]. As a repair material, lime is widely used to improve the mechanical properties of rammed soils in the restoration of ancient rammed earth projects [3,4,5,6,7,8,9,10,11]. However, due to the differences in external environment, material composition and construction technology [12,13,14,15,16,17], a large number of cracks, flakes and other diseases often appear on the outer surface of earthen city walls repaired with lime soil. As shown in Figure 1, the quality of the newly built earthen city wall is inferior to that of the old original city wall. The question of how to avoid and improve the diseases of the newly built city wall has attracted the attention of the cultural relic community; it is known that the stable reaction of lime and the optimization of lime soil construction technology are the most critical links.

The stable reaction of the lime soil has been widely investigated. As a chemical-physical improvement method, lime can significantly improve the mechanical properties of soil through pozzolanic reactions, carbonation reactions, cation exchange and crystallization [18,19,20,21,22,23,24]. Soil type, type and amount of lime added, curing period and method, moisture content, method of compaction, and time elapsed between mixing and compaction influence both the short-term in cation exchange and flocculation, agglomeration and long-term in the formation of cementitious gels by pozzolanic reaction with the reaction mechanism of lime stabilization [25,26,27,28].

In order to stabilize the reaction of lime, improve the performance of lime soil and optimize the construction technology, scholars across the world have carried out a lot of research on the “lime slaking” of lime soil. Gu et al. [29] found that the temperature of lime digestion water, the mass ratio of water to lime and the slaking time had an effect on the activity of calcium hydroxide. It was concluded that the dispersion of Ca(OH)_2_ has better dispersion and the higher activity when the digestion temperature was 84 °C, the mass ratio of water to lime was 5:1 and unslacking was performed. Wei et al. [30] concluded that the nanoparticle size and high reactivity of lime improve the properties such as compressive strength and surface hardness after slaking. Yue et al. [31] determined that the volume crack and expansion shrinkage rates of the sample after the dry wet cycle initially decreased, and then stabilized with increasing aging time. Different authors have analyzed the change in physical and microstructural characteristics of lime putty with slaking time [32,33,34,35,36,37]. The microstructure evolution confirms that the shape of Portland Ca(OH)_2_ prismatic crystal changes to plate crystal, which improves the plasticity of putty. Rosell et al. [38] concluded that the amount of water used and the temperature reached during slaking affect the particle size distribution of the lime putties. In addition, lime putty experiences a significant increase in viscosity in the slaking process [39]. In fact, the construction process and operation and maintenance process of lime soil walls are long-term carbonization reaction processes [40,41]. Due to the external environment and carbonization expansion, the external surface of lime soil walls suffers from surface cracks, warping and peeling [42,43,44]. If the lime soil is properly matured in the early stage and used for the city wall, it raises questions regarding what the changes in the surface and performance of the city wall will be. It is worth noting that the particle gradation and pH value of lime soils under different slaking conditions deserve attention.

To explore the influence of slaking on the surface cracks and mechanical properties of the mended lime soil of the Kaifeng city wall, lime soil was prepared according to the mix proportions of the mended lime soil. Combined with the engineering construction technology, tests were designed considering the slaking time, lime content and moisture content. The influence of these factors on surface cracks, mechanical properties, particle gradation, pH value and microstructure was studied and the reasonable composition and slaking time of lime soil materials for wall repair in the Kaifeng area were determined.

## 2. Materials and Methods

### 2.1. Materials

The soil used in the test was obtained from the peeling of a soil site in Kaifeng City. The soil is silty clay, with unfavorable characteristics, such as loose structure, low strength, strong capillarity and poor water stability. The tests also show that the natural moisture content is 13.20%, the plastic limit is 21.03%, the liquid limit is 32%, the plasticity index is 16.6, the optimum moisture content is 14.32% and the maximum dry density is 1.68 g/cm^3^. The physical property indexes are shown in Table 1. The composition of silty clay is analyzed qualitatively and quantitatively by X-ray diffraction (Bruker-AXS Company, Karlsruhe, Germany). The oxide content is shown in Table 2. High active CaO used for lime soil is large and white in appearance. In addition, lime is in powder form with the particle size of 150–200 mesh.

### 2.2. Methods

The effects and mechanism of the slaking time, lime content, and moisture content on the cracking and mechanical properties of lime soil were analyzed by the test results of surface cracks, triaxial compression, particle gradation, pH value, X-ray diffraction and scanning electron microscope.

#### 2.2.1. Experimental Factors

A series of tests were carried out to explore the deterioration mechanisms of surface cracks and mechanical properties of lime soil with slaking time, lime content and moisture content. The tests include surface cracks, triaxial compression, particle gradation, pH value, X-ray diffraction and scanning electron microscope. According to the relevant literature [45,46,47], the slaking times of the lime soil (T) are taken as 0 h, 12 h, 24 h, 48 h and 72 h, the lime content (M) is 10%, 15% and 20%, and the moisture content (W) is 14%, 18% and 26%, respectively, considering the characteristics of lime soil of the city wall and time cost (Table 3).

#### 2.2.2. Lime Soil Slaking

Experiments on the slaking time, lime content and moisture content of specimens were conducted according to the Standard for Geotechnical Engineering testing method (GB/T50123-2019) [48]. The dry soil after 2 mm sieving was taken and mixed thoroughly with lime and water, then placed in a constant temperature and humidity box for slaking. The temperature of the programmable constant temperature and humidity test chamber (Dongguan Baker Testing Equipment Limited Company, Guangdong, China) was set to 20 ± 2 °C and the humidity was set to 95 ± 1%. It was added 10 g of water every 12 h.

#### 2.2.3. Sieve Analysis of Particle Gradation

To observe the change in particle gradation of lime soil during slaking, 600 g of dried specimens (divided into three parts) were taken every 12 h for sieve analysis. The fine screen apertures were 2 mm, 1 mm, 0.5 mm, 0.25 mm, 0.1 mm and 0.075 mm, respectively. The numerical error was controlled within ±1 g.

#### 2.2.4. The pH Tests

The pH5S (Shanghai Sanxin Instrument Factory, Shanghai, China) puncture pen pH meter is used for pH detection. The pH5S puncture electrode is inserted into 7.00 pH, 4.00 pH and 10.01 pH solutions in turn for 3-point calibration, then inserted into three different positions of lime soil for pH detection, and the pH value error shall be controlled within ±0.05. The results are taken as the average value. To observe the pH change in lime soil during slaking, the pH is measured every 1 h for the first 24 h and every 4 h for the next 48 h.

#### 2.2.5. Surface Cracks

The slaked lime soil specimens must be photographed at regular intervals in the natural environment. Each image is processed by Photoshop, and the crack ratio (Figure 2) is calculated by the particles (Pores) and cracks analysis system (PCAS). The quantification of the crack image involves the following three steps: image segmentation, crack identification and measurement [49]. The slaked lime soil specimens are 61.8 mm in diameter and 125 mm in height. The photographing times are as follows: 0 h, 0.5 h, 1 h, 2 h, 3 h, 4 h, 5 h, 6 h, respectively.

#### 2.2.6. Triaxial Compression Test under Low Confining Pressure

Tsz-10 (Nanjing Ningxi soil Instrument Limited Company, Nanjing, China) strain-controlled triaxial apparatus was used in the triaxial compression experiments. Considering the actual confining pressure conditions, the low confining pressures were set to 50 kPa, 100 kPa and 150 kPa. The shear rate was 0.8%/min. The lime soil was slaked for 0 h, 12 h, 24 h, 48 h, and 72 h to produce specimens of 39.1 mm in diameter and 80 mm in height. The specimens were put into the triaxial apparatus and subjected to unconsolidated undrained tests at different low confining pressures.

#### 2.2.7. X-ray Diffraction

The 2 g specimens with different slaking times were dried and ground to powder. The phase composition of the specimens was tested by Bruker D8 Advance’s X-ray powder diffractometer (Bruker-AXS Company, Karlsruhe, Germany). The instrument parameters were as follows: ceramic X-ray tube, Cu target, power 2.2 kw. The angle scanning range (2θ) was 20~70°, the scanning speed was 5°/min, and the step interval was 0.02°.

#### 2.2.8. Scanning Electron Micrographs

After the triaxial compression test, 2 g specimens were taken into a drying oven and dried at 105 °C for 12 h. The specimens were then put into a coating machine, and platinum was sprayed on the surface of the specimens. The micro morphology of the specimen fracture was observed with an FEI QUANTA 250 (FEI Company, Hillsboro, OR, USA) environmental field emission scanning electron microscope. The instrument parameters were as follows: secondary electron image, high vacuum mode, test voltage of 1 kV and test current of 10 mA.

## 3. Results

### 3.1. Particle Gradation

Figure 3 shows the particle gradation curves of lime soil with different slaking times, lime contents and moisture contents. It can be observed from the Figure 3a–c that the particle grading curves of 10%, 15% and 20% lime soil vary regularly with the increase in slaking time. With the increase in slaking time, the particle grading curves of lime soil with different contents first decreases and then tends to be flat or approximately increases. Figure 3d shows that the particle gradation curves of the specimens decrease first and then increase with the increase in water content. This is because lime reacts rapidly in the presence of water after mixing with soil. Ca(OH)_2_ reacts with CO_2_ to form CaCO_3_ crystals, which form a cohesive polymer through bonding. The attraction between particles increases, resulting in flocculation and coagulation [50,51], which affects the particle gradation composition of the soil samples and consequently reduces the particle gradation curves.

From the changes in particle size and composition of lime soil under different slaking time, it can be divided into two aspects. On the one hand, the mass percentage of lime soil with particle sizes less than 2 mm, 1 mm, 0.5 mm and 0.25 mm gradually decreases with the increase in slaking time. On the other hand, the mass percentage of lime soil with particle sizes less than 0.1 mm and 0.074 mm increases after 12 h of slaking. When lime soil continues to be slaked, the percentage of mass with particle size less than 0.1 mm and 0.074 mm decreases. Flocculation and coagulation between particles increase during lime soil slaking, gradually increasing the percentage of mass with particle sizes greater than 0.25 mm and 2 mm. Although the addition of lime changes the composition of the particles, the lime destroys the connection between soil particles during the slaking process, producing more small particles with particle sizes less than 0.1 mm and 0.074 mm. Compared with 0 h of slaking, after 12 h of slaking, the mass percentages of particles with particle sizes less than 0.1 mm increases by 3.65%, 2.94% and 15.05% and the mass percentage of particles with particle sizes less than 0.075mm decreases by 2.72%, 1.48%, and 14.04%, respectively. At this time, the particle grading curves of the soil specimens evidently tend to be stable, indicating that the reaction rate of lime slows down and the particle influence on size composition of soil specimens slows down.

### 3.2. Analysis of pH Test Results

The relationship between the pH values of the specimens with different lime contents and slaking times is shown in Figure 4. It can be observed from the figure that with the increase in slaking time, the pH value of the specimens with different lime contents first increases and then decreases. It shows that the more lime that is added, the longer the slaking time required for the ash soil to reach the peak pH value, and the greater the peak pH value. For 10% lime soil, a peak pH value of 13.47 appears after 25 h of slaking; for 15% lime soil, a peak pH value of 13.61 appears after 25 h of slaking; and for the 15% lime soil, the peak pH value of 13.63 appears after 30 h of slaking. The relationship between the pH values of the specimens with different moisture contents is shown in Figure 5. It shows that the pH value of the sample increases with the increase in moisture content. The more lime and water that are added, the longer lime is involved in the reaction. The longer the slaking time is required to reach the peak of pH of a high content lime soil, the more OH^−^ is produced in the slaking reaction.

During slaking, the clay lime mixture begins to react upon the addition of water. The hydrated lime is decomposed into Ca^2+^ and 2(OH)^−^, and the pH of the mixture increases. CO_2_ in the air participates in the reaction and consumes OH^−^, resulting in a decrease in the pH value of the mixture [52]. In practical projects, workers can control the pH value to master the slaking state of lime soil, and thus determine the optimal construction time. Comparing Figure 3, Figure 4 and Figure 5, it can be found that the pH value approaches the peak and the particle grading curve tends to be stable after 24 h of slaking. Considering the requirements of saving time in actual construction, the recommended value of slaking time is 24 h.

### 3.3. Surface Cracks

The surface cracks and crack ratios of specimens with different slaking times, lime contents and moisture contents under natural conditions are shown in Figure 6 and Table 4, respectively. Under the natural environment, cracks appear at 0.5 h for the unslaked specimens (0 h), and the cracks gradually increase with the passage of time. The surface cracks of specimens with different contents of lime soil are significantly reduced after 12 h, 24 h, 48 h and 72 h of slaking, compared with that of unslaked specimens (0 h). After 6 h of natural curing, the crack ratios are 6.09%, 6.81% and 13.63% for lime soil specimens (0 h) with 10%, 15% and 20% content, respectively. However, the crack ratios are 0.11%, 0.21% and 0.39% for lime soil specimens (12 h) with 10%, 15% and 20% content, respectively. Proper slaking of lime soil can reduce the surface cracks of specimens and improve the durability. Combined with practical engineering, the experimental phenomena are very similar to the cracks, spalling and other diseases of the newly built Kaifeng city wall at present. In the natural environment, the surface cracks of unslaked lime soil specimens increase significantly with time and finally lose their durability.

The surface cracks of the lime soil specimens are also closely related to the content of lime and water. The occurrence of surface cracks in lime soil specimens is directly proportional to the content of lime and inversely proportional to the moisture content. Compared with 10% lime content (0 h), the crack ratios are increased by 11.82% and 123.81% for the specimens (0 h) with 15% and 20% content after 6 h of natural curing, respectively. Compared with 14% moisture content (0 h), the crack ratios are decreased by 77.89% and 80.93% for the specimens (0 h) with 18% and 22% content after 2 h of natural curing, respectively. In the process of natural curing, the increase in lime content and the decrease in moisture content will accelerate the chemical reaction rate of the specimens, produce more surface cracks and reduce their durability.

Combined with the results of particle gradation and the pH value of lime soil under different slaking times, lime contents and moisture contents, the reasons for the significant increase in surface cracks over time in unslaked lime soil specimens in natural environment can be attributed to the following: (1) the cracks are caused by chemical reaction of lime soil, and the increase in slaking time, lime content and moisture content will accelerate the chemical reaction rate of the specimen. During the experiment, the surface cracks of the unslaked lime soil specimens increase with time in the natural environment. At the same time, white substance precipitation and water loss are observed on many different parts of the specimen surface. It is considered that Ca(OH)_2_ generated by the reaction of quicklime with water in lime soil reacts with CO_2_ in the air to generate white CaCO_3_ material. If the reaction continues after the lime soil hardens, it will expand due to the continuous reaction of the lime, resulting in uplift and cracking [30]. (2) The other reason is particle recombination and strong alkaline environment. The amount of lime in lime soil provides the specimen with a high pH environment. At a high pH value, the particles of the specimen will reorganize over time, destroying the original particle gradation of the soil specimen and producing more small particles. The generation of these small particles will fill and squeeze the specimens, and even expand, further accelerating the creation of cracks. The reason for the reduction in cracks in the lime soil specimens slaked for 12 h is that slaking gives lime some time for violent reactions and particle reorganization. After 12 h, the reaction of lime soil tends to level off, thus, reducing the surface cracks of the specimens. A certain amount of slaking time provides quicklime with sufficient time for curing, carbonization reactions and particle reorganization, thus, reducing the possibility of macroscopic cracks in lime soil due to continuous carbonization, and thus improving the durability of lime soil in repairing the city walls.

### 3.4. Analysis of Mechanical Properties

#### 3.4.1. Stress–Strain Curve

The principal stress difference axial strain curves of lime soil with different lime contents with different slaking times and moisture contents are shown in Figure 7. The slaking time, lime content, moisture content and net confining pressure all affect the stress–strain curves of lime soil. With the increase in slaking time, the peak of the stress–strain curve of lime soil decreases after it first rises, and the peak value appears at 12 h of specimen slaking; with the increase in lime content, confining pressure and the decrease in moisture content, the peak value of the stress–strain curve of lime soil increases gradually. The strength of the specimens with 10% ash content increases by 12.84%, 8.55% and 3.86% after 12 h of proper slaking at the confining pressures of 50 kPa, 100 kPa and 150 kPa, respectively; the strength of the specimens with 15% ash content increases by 7.24%, 10.49% and 6.03%; the strength of the specimens with 20% ash content increases by 2.14%, 3.96% and 7.17%. Under an environment of high pH value, the lime soil has a relatively uniform particle gradation distribution after 12 h of proper slaking, which plays a role in filling, bonding and curing soil particles. Therefore, proper slaking of lime soil can increase the peak value of the stress–strain curve and accordingly improve the strength.

The figures show that the larger the lime content, the higher the confining pressure and the smaller the moisture content, the greater the peak value of the stress–strain curve and the more obvious the softening characteristics. On the contrary, the less lime content, the lower the confining pressure and the greater the moisture content, the lower the peak value of the stress–strain curve and the more obvious the strain hardening characteristics. When the lime content is 20% and the moisture content is 14%, the stress–strain curves under low confining pressures show strain softening characteristics. The particle size of quicklime powder selected in the experiment is 150–200 mesh, which can be used as “fine aggregate” in the silty clay to fill, bond and solidify soil particles and enhance the soil strength. The increase in lime content and the decrease in moisture content can enhance the bonding effect and improve the strength of the samples. However, due to the high lime content and the small moisture content, uneven particle gradation, brittle failure easily occurs, resulting in the softening of the stress–strain curve.

#### 3.4.2. Shear Index

The relationship between the cohesion and internal friction angle of the specimens with different slaking times, lime contents and moisture contents is shown in Figure 8. With the increase in slaking time, the cohesion of the specimens with different lime contents first increases and then decreases, while the internal friction angle first decreases and then increases. Compared to other slaking times, the shear index of lime soil slaked for 12 h is better. However, the longer the lime soil slaking time is, the more unfavorable the shear index is. Compared to the unslaked specimens, the cohesion of the specimens with 10%, 15% and 20% lime contents after 12 h of slaking increases by 41.88%, 9.10% and 20.27%, respectively and the internal friction angle decreases by 3.81%, 1.18% and 11.27%, respectively. After 72 h of slaking, the cohesion of the specimens with 10%, 15% and 20% lime content increases by 7.44%, decreases by 15.05% and 8.21%, respectively, and the internal friction angle increases by 2.97%, 7.44% and 2.82%, respectively. It is worth noting that the long-term slaking process has a negative impact on the shear index of lime soil. Controlling the reasonable slaking time of lime soil is important to improve the mechanical properties of lime soil. In industrial production, it is necessary to digest the lime completely in order to avoid alkali return. In order to digest the lime completely and avoid problems, such as alkali return, the slaking time is determined to be about 12 h in industrial production [29]. Therefore, it is suggested that the slaking time of lime soil is taken as 12 h.

With the increase in lime content and the decrease in moisture content, the cohesion and internal friction angle of the specimens increase. Compared with 10% lime, the cohesion of the specimens with 15% and 20% lime after 0 h slaking increased by 95.23%, and 188.42%, and the internal friction angle decreased by 0.21% and 1.97%, respectively; Compared with the moisture content of 22%, the cohesion of the specimens with moisture contents of 14% and 18% increased by 68.24% and 24.60% and the internal friction angle decreased by 47.50% and 13.08%, respectively. Although the shear index of the specimens with high lime content and low moisture content is good, the specimens exhibited many cracks and decreased in durability. Comparing Figure 6 and Figure 8, a reasonable slaking time of 12 h, lime content, and moisture content can improve the mechanical properties of the specimens and improve the surface cracks. Therefore, it is recommended to repair the wall soil after the lime soil is properly slaked for 12 hours under high water content before construction.

### 3.5. Micro Mechanism Analysis

Figure 9 is the XRD diagram of different lime contents slaked for 0 h, 24 h, 48 h and 72 h, respectively. Compared with the unslaked specimens, the diffraction peak of slaked CaO decreased, the diffraction peak of Ca(OH)_2_ increased first and then decreased, and the diffraction peak of calcite increased. During slaking, some active CaO in lime soil is hydrated to produce Ca(OH)_2_. The reaction formula is as follows (1). The increase in Ca(OH)_2_ content during lime soil slaking increases the pH value. In an environment with a high pH value, the particle size composition, surface crack and mechanical properties of lime soil are improved in the slaking process. The lime soil continues to slake, and Ca(OH)_2_ reacts with CO_2_ in the air to form CaCO_3_. The reaction formula is as shown in (2). This reaction will consume part of the OH− generated by the reaction of CaO with water. Therefore, the content of Ca(OH)_2_ decreased and the content of CaCO_3_ increased continuously. As the slaking time of lime soil increases and the pH value decreases, the particle gradation becomes worse and the mechanical properties decrease. Lime provides lime soil with a strong alkaline environment that hydrates the clay minerals silicates and aluminates [53] under the action of calcium ions. Calcium carbonate crystals formed by the carbonation of low calcium silicate and CO_2_ and highly polymerized amorphous silica gel are the source of strength growth of hardened bodies [54]. The cementitious material generated during the reaction of lime soil can effectively improve the local expansion and cracks caused by the lime curing and carbonization reaction of lime in the curing process of unslaked lime soil, and improve the mechanical properties of lime soil accordingly.
CaO + H_2_O = Ca(OH)_2_(1)
Ca(OH)_2_ + CO_2_ + H_2_O = CaCO_3_↓ + 2H_2_O(2)

The specimens were magnified 2000 times by a scanning electron microscope to observe the particle morphology and pore characteristics of the lime soil specimens (Figure 10). The 20% lime soil samples with different slaking times were taken for analysis to further explore the influence of the mechanism of lime slaking time on the surface cracks and mechanical properties of wall repair soil. In the process of lime soil slaking, the flocs generated by the hydration and carbonization of lime in the specimen not only fill the gap between the soil particles, but also enhance the cementation between soil particles, gradually forming a dense whole, and effectively improve the mechanical properties. It can be observed from Figure 10b that the high pH environment of the unslaked lime soil destroys the original microstructure of the clay [55], resulting in a large number of small cracks. The large particles are decomposed into many small particles, and the small particles gather together to further break the clay particles into smaller particles [56], thus forming the cracks on the outer surface of the unslaked lime soil specimen in Figure 6. It can be observed from Figure 10b that after 12 h of slaking, the ash soil has a uniform soil particle arrangement, good particle grading, a clear reduction in specimen curing surface cracks, more adhesion on the surface of soil particles, improved adhesion between the particles and improved mechanical properties. If the lime soil slaking time is too long, the internal pores of the specimen increase, the soil particles are unevenly arranged, the particle grading is poor, the adhesion between particles is weakened, and the mechanical properties are reduced. During the slaking process of lime soil, Ca(OH)_2_ and CaCO_3_ crystals generated by various reactions, such as hydration and carbonization of lime, “grow” in the pores of soil particles, showing an irregular flake crystal growth structure, forming a bridge structure between the soil particles, promoting the bonding of the dispersed soil particles into a whole, and effectively improving the mechanical properties of lime soil.

## 4. Discussion

Figure 11 shows the relationship between particle curve, pH value, crack ratio, cohesion and internal friction angle of the specimen. Combined with the analysis of Figure 11 and microstructure, the purpose is to explore the internal mechanism of slaking that affects the mechanical properties of lime soil. It can be observed from the figure that with the increase in slaking time, the crack ratio of the specimen decreases, the cohesion first increases and then decreases, and the internal friction angle first decreases and then increases. The more lime and less water, the stronger the crack ratio and mechanical properties of the lime soil. For this experimental phenomenon, the pH and particle gradation of lime soil under different slaking times, lime contents, and moisture contents are tested to find the internal mechanism. From the experimental results, the pH value of lime soil first increases and then decreases with the increase in slaking time, the pH value increases with the increase in lime content and moisture content, and the pH value is greater than 12.5. In a high pH environment, slaking will reorganize the particle composition of lime soil, so that the percentage of mass with particle sizes less than 2 mm gradually decreases, and the percentage of mass with particle sizes less than 0.074 mm first increases and then decreases or tends to be flat. In the slaking process, the Ca(OH)_2_ substance produced by the reaction of lime and water endows lime soil with extremely high strong alkalinity. Lime with the particle size of 150–200 mesh acts as the “fine aggregate” in silty clay and plays a bonding role in the slaking process of lime soil, which makes some lime and soil particles form a cohesive body, increasing the particles with particle sizes greater than 2 mm and reducing the particles with particle sizes less than 0.074 mm. The microstructure of lime soil before and after slaking shows that the reaction process of lime soil has the ability to “destroy” silty clay particles on the one hand, and the generation of cementitious materials on the other hand. Therefore, the slaking process of lime soil involves the generation of cemented large particles and the reproduction of small particles. This helps to explain why the cohesion of lime soil has increased after slaking (12 h), and the internal friction angle has decreased. However, the number of large and small particles bonded in the lime soil slaked for a long time tends to be stable, the cementitious material of the specimen is limited, and the bonding performance decreases, resulting in a decrease in the cohesion of the specimen and an increase in the internal friction angle.

## 5. Conclusions

Based on the principles of slaking reactions and carbonization, tests on lime soil with different slaking conditions confirmed that the slaking time can improve the surface cracks and mechanical properties of the specimens. The main conclusions are summarized as follows.

With the increase in slaking time, the crack ratio of the specimen decreases, the cohesion first increases and then decreases, and the internal friction angle first decreases and then increases. The particle curve of lime soil shows an overall downward trend; the pH value first increases and then decreases. Proper slaking time (12 h) of lime soil can accelerate the reaction rate of lime in a high alkali environment, improve particle gradation, further reduce surface cracks and improve mechanical properties. However, it is worth noting that the long-term slaking process has a negative impact on the shear index of lime soil.The less lime and more water, the less the specimen crack ratio. The more lime and less water, the stronger the mechanical properties of lime soil. The more lime and water, the more the number of particles with particle size less than 2 mm, and the fewer particles of sizes less than 0.074 mm, the greater the peak value. In order to avoid the surface diseases caused by the chemical reaction of lime soil and improve the durability, it is suggested that the specimen should have a high moisture content and be slaked for 12 h in practical engineering.After the lime soil is slaked for 12 h, the particles less than 0.075 mm increase and the pH value increases to 13. Under a high pH environment, the unslaked lime causes the crushing of clay particles, and the resulting fine cracks explain the cracks on the outer surface of lime soil specimens. Ca(OH)_2_, CaCO_3_, and other cementitious substances produced by reasonably slaking lime soil can improve the crushing of clay particles in a high pH environment, form aggregates, fill the pores between the specimen particles and effectively improve the mechanical properties.Proper slaking of lime soil for 12 h and a high moisture content can accelerate the reaction speed of lime, reduce the damage to specimen particles in a high pH environment, improve particle gradation, further reduce surface cracks and improve mechanical properties. By controlling the slaking time and moisture content of lime soil, the durability and surface performance of the repaired city wall can be improved, which is of great significance to the restoration of Kaifeng city wall.

In addition, the particle size, carbonization condition, and carbonization time of lime will affect the cracking and mechanical properties of the material, which need to be further studied.

## Figures and Tables

**Figure 1 materials-15-04151-f001:**
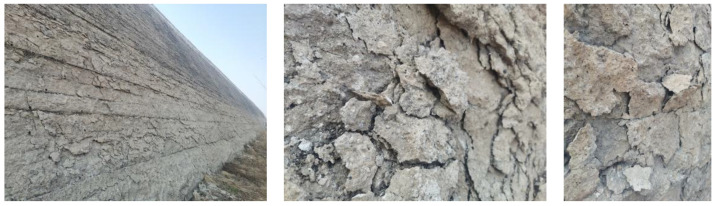
Diseases of the newly repaired earthen city walls.

**Figure 2 materials-15-04151-f002:**
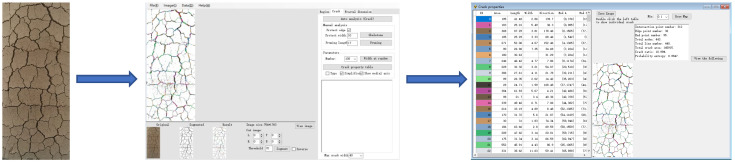
The quantification of the crack image (PCAS).

**Figure 3 materials-15-04151-f003:**
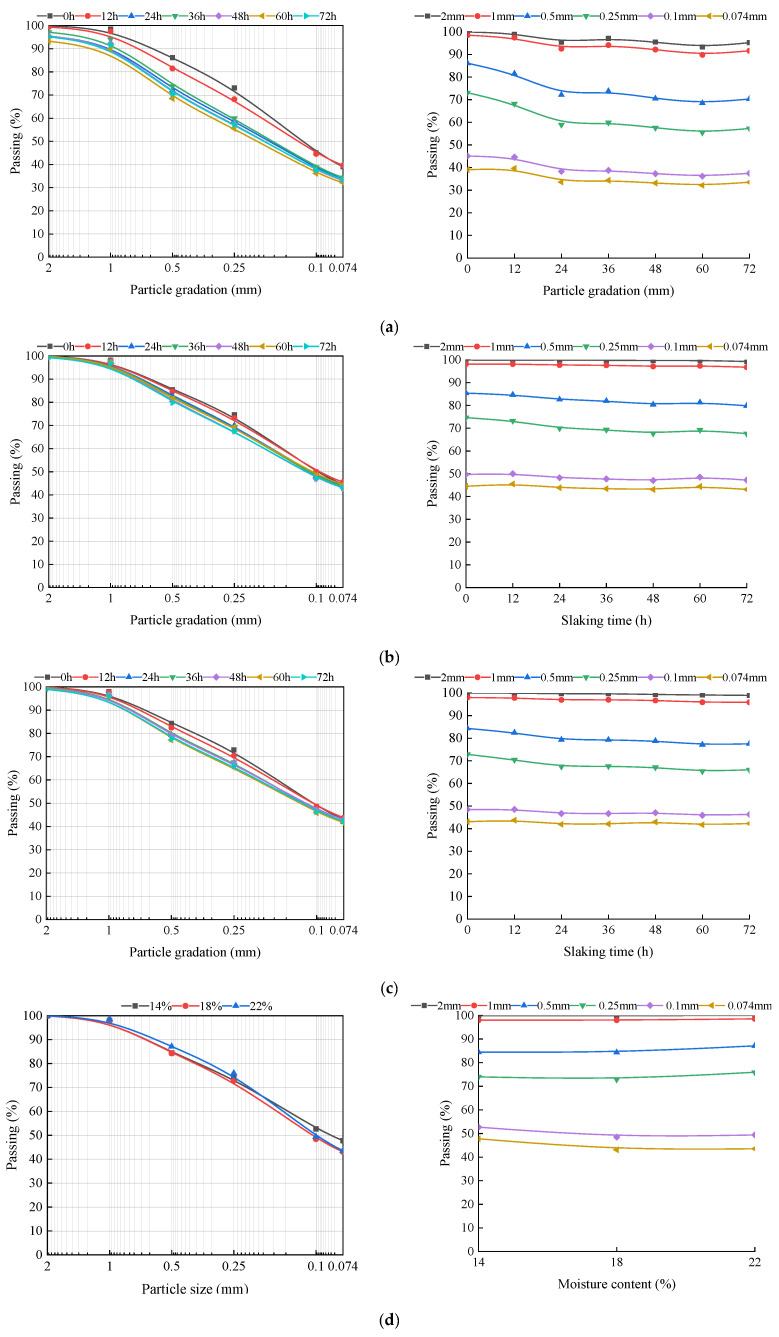
Particle gradation of specimens with different slaking times, lime contents and moisture contents: (**a**) M = 10%, W = 18%; (**b**) M = 15%, W = 18%; (**c**) M = 20%, W = 18%; (**d**) T = 0 h, M = 20%.

**Figure 4 materials-15-04151-f004:**
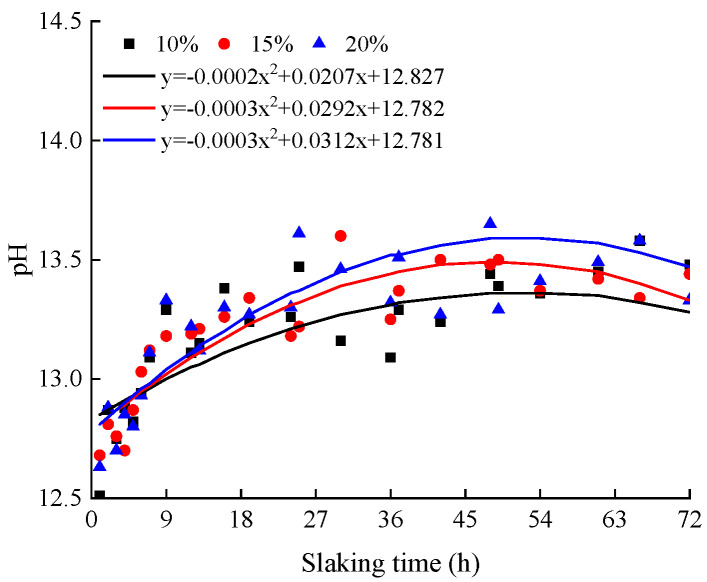
The pH curves of lime soils with different slaking times.

**Figure 5 materials-15-04151-f005:**
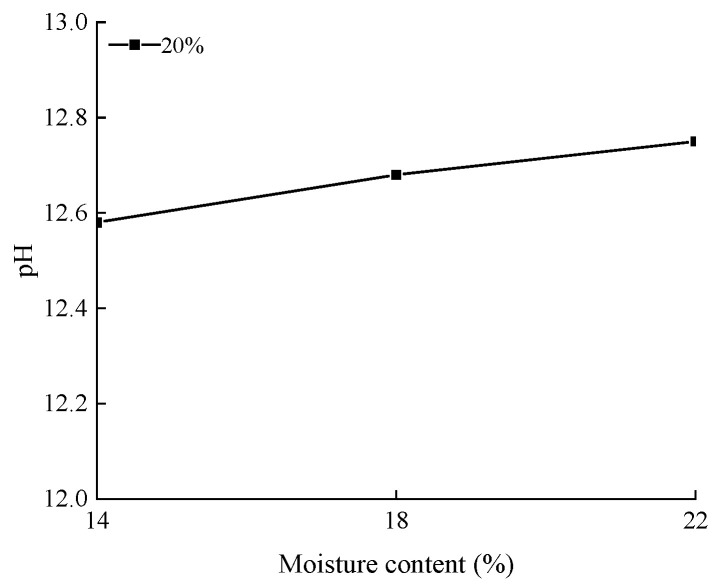
The pH curves of lime soils with different moisture contents.

**Figure 6 materials-15-04151-f006:**
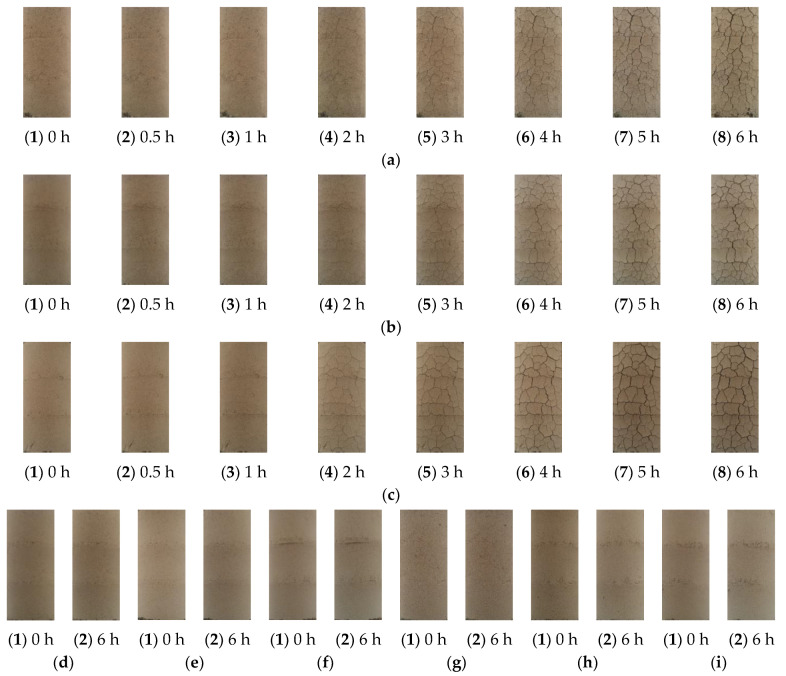
Surface cracks in unslaked and slaked specimens: (**a**) T = 0 h, M = 10%, W = 18%; (**b**) T = 0 h, M = 15%, W = 18%; (**c**) T = 0 h, M = 20%, W = 18%; (**d**) T = 12 h, M = 10%, W = 18%; (**e**) T = 12 h, M = 15%, W = 18%; (**f**) T = 12 h, M = 20%, W = 18%; (**g**) T = 24 h, M = 10%, W = 18%; (**h**) T = 24 h, M = 15%, W = 18%; (**i**) T = 24 h, M = 20%, W = 18%; (**j**) T = 48 h, M = 20%, W = 18%; (**k**) T = 48 h, M = 15%, W = 18%; (**l**) T = 48 h, M = 20%, W = 18%; (**m**) T = 72 h, M = 10%, W = 18%; (**n**) T = 72 h, M = 15%, W = 18%; (**o**) T = 72 h, M = 20%, W = 18%; (**p**) T = 0 h, M = 20%, W = 14%; (**q**) T = 0 h, M = 20%, W = 22%.

**Figure 7 materials-15-04151-f007:**
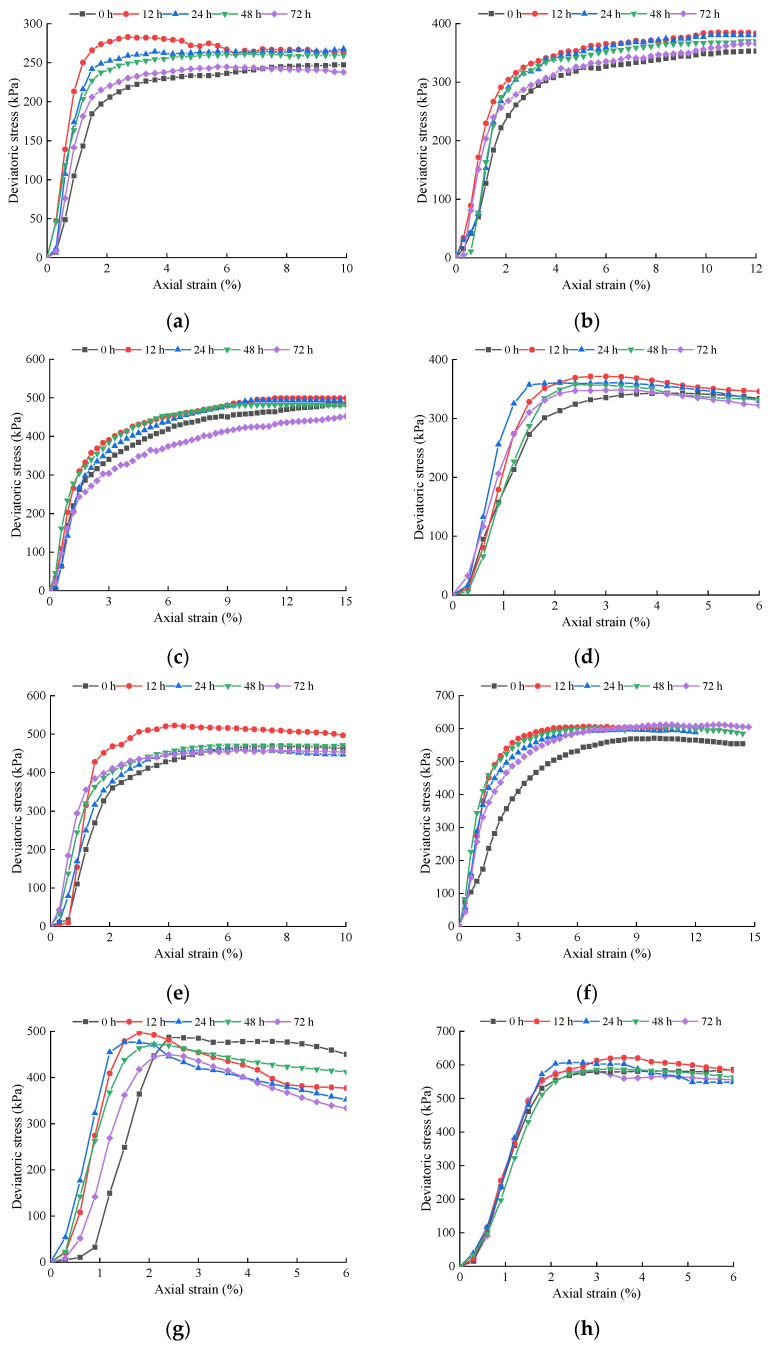
Principal stress difference axial strain curves of lime soil: (**a**) M = 10%, W = 18%, 50 kPa; (**b**) M = 10%, W = 18%, 100 kPa; (**c**) M = 10%, W = 18%, 150 kPa; (**d**) M = 15%, W = 18%, 50 kPa; (**e**) M = 15%, 100 kPa; (**f**) M = 15%, W = 18%, 150 kPa; (**g**) M = 20%, W = 18%, 50 kPa; (**h**) M = 20%, W = 18%, 100 kPa; (**i**) M = 20%, W = 18%, 150 kPa; (**j**) T = 0 h, M = 20%.

**Figure 8 materials-15-04151-f008:**
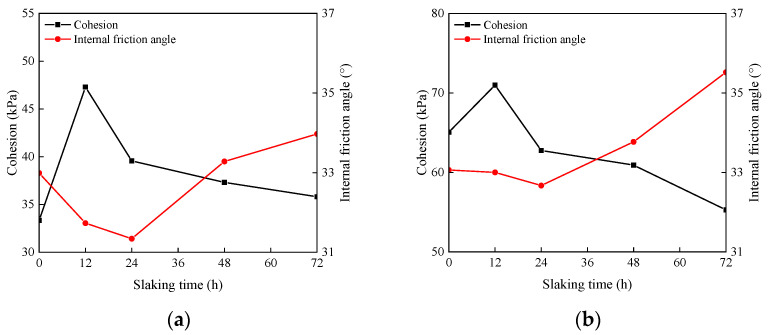
Relation curve between cohesion and internal friction angle of specimens: (**a**) M = 10%, W = 18%; (**b**) M = 15%, W = 18%; (**c**) M = 20%, W = 18%; (**d**) T = 0 h, M = 20%.

**Figure 9 materials-15-04151-f009:**
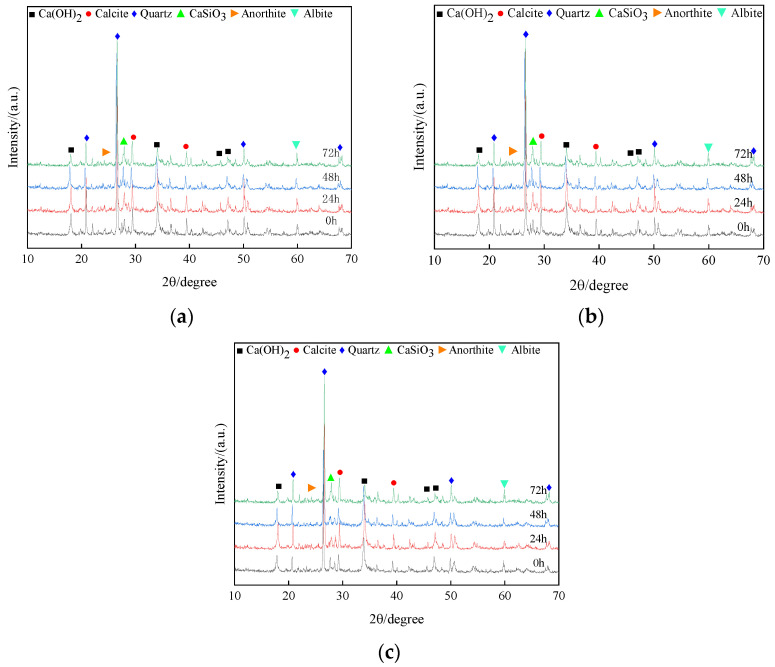
XRD of different amounts of lime slaked for 0 h, 24 h, 48 h and 72 h under standard curing: (**a**) M = 10%; (**b**) M = 15%; (**c**) M = 20%.

**Figure 10 materials-15-04151-f010:**
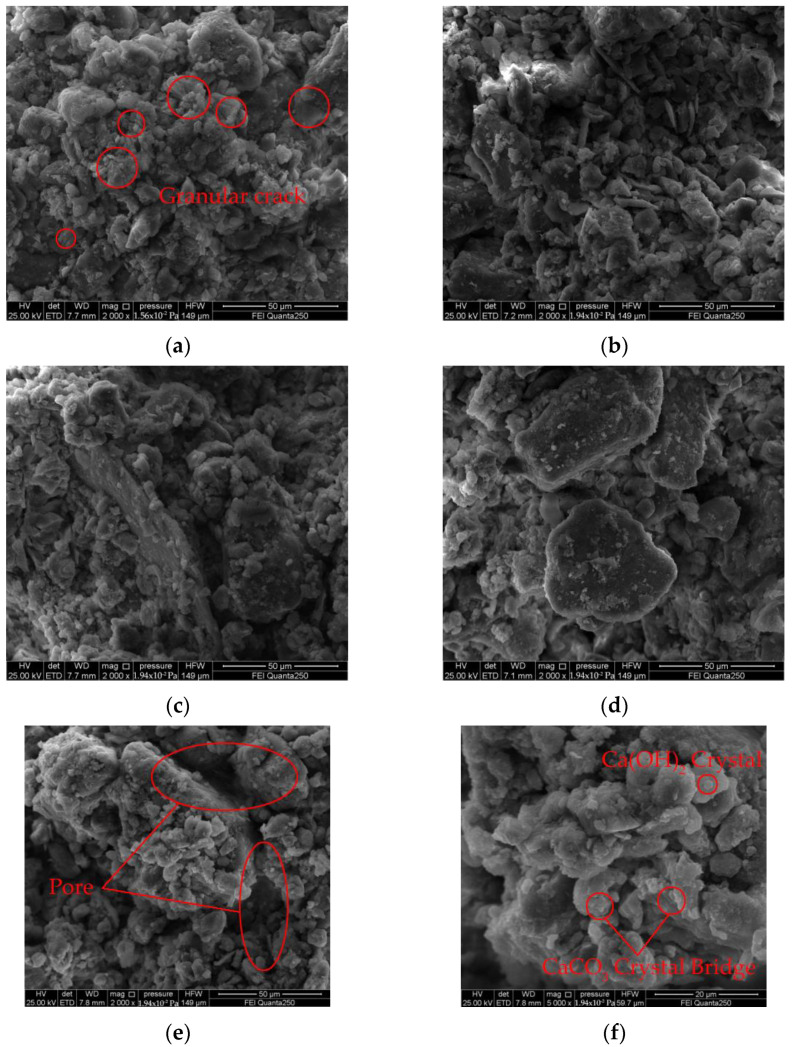
Scanning electron microscope: (**a**) T = 0 h, M = 20%; (**b**) T = 12 h, M = 20%; (**c**) T = 24 h, M = 20%; (**d**) T = 48 h, M = 20%; (**e**) T = 72 h, M = 20%; (**f**) T = 72 h, M = 20%.

**Figure 11 materials-15-04151-f011:**
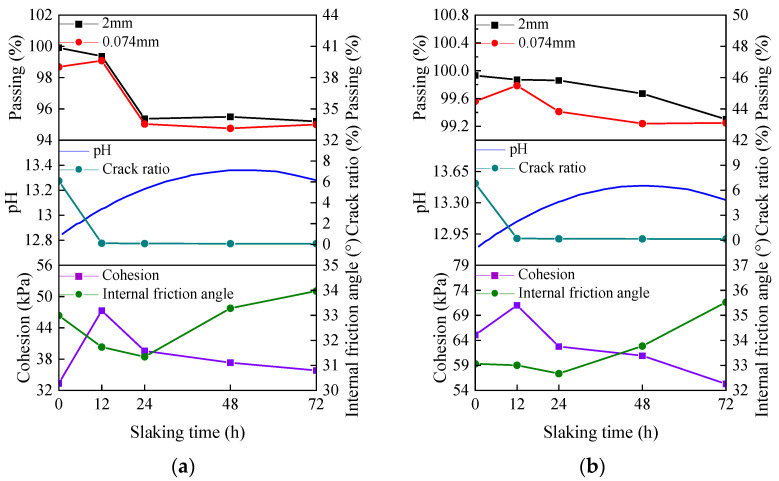
The relationship curve particle curve, pH value, crack ratio, cohesion and internal friction angle of specimen: (**a**) M = 10%, W = 18%; (**b**) M = 15%, W = 18%; (**c**) M = 20%, W = 18%; (**d**) T = 0 h, M = 20%.

**Table 1 materials-15-04151-t001:** Physical characteristics of soil.

Natural Moisture Content (%)	Plastic Limit W_P_ (%)	Liquid Limit W_L_(%)	Plasticity Index	Optimum Moisture Content (%)	Maximum Dry Density (g/cm^3^)	Particle Composition (%)					
						<2 mm	<1 mm	<0.5 mm	<0.25 mm	<0.1 mm	<0.074 mm
13.20	21.03	37.63	16.6	14.32	1.68	95.87	92.40	70.51	56.59	38.62	35.67

**Table 2 materials-15-04151-t002:** Chemical composition of soil (wt.%).

SiO_2_	CaO	Al_2_O_3_	MgO	Fe_2_O_3_	K_2_O	Na_2_O	∑
61.79	8.91	15.83	2.67	5.23	2.88	1.48	98.79

**Table 3 materials-15-04151-t003:** Experimental factors of lime soil.

Grouping	Slaking Time (h)	Lime Content (%)	Moisture Content (%)
T = 0 h, M = 10%, W = 18%	0	10	18
T = 0 h, M = 15%, W = 18%		15	
T = 0 h, M = 20%, W = 18%		20	
T = 12 h, M = 10%, W = 18%	12	10	18
T = 12 h, M = 15%, W = 18%		15	
T = 12 h, M = 20%, W = 18%		20	
T = 24 h, M = 10%, W = 18%	24	10	18
T = 24 h, M = 15%, W = 18%		15	
T = 24 h, M = 20%, W = 18%		20	
T = 48 h, M = 10%, W = 18%	48	10	18
T = 48 h, M = 15%, W = 18%		15	
T = 48 h, M = 20%, W = 18%		20	
T = 72 h, M = 10%, W = 18%	72	10	18
T = 72 h, M = 15%, W = 18%		15	
T = 72 h, M = 20%, W = 18%		20	
T = 0 h, M = 20%, W = 14%	0	20	14
T = 0 h, M = 20%, W = 22%	0	20	22

**Table 4 materials-15-04151-t004:** The crack ratios of specimens with different slaking conditions.

Grouping	Time						
	0.5 h	1 h	2 h	3 h	4 h	5 h	6 h
T = 0 h, M = 10%, W = 18%	0.15%	0.08%	2.39%	3.00%	3.99%	5.19%	6.09%
T = 0 h, M = 15%, W = 18%	0.81%	0.95%	2.35%	3.40%	4.56%	5.99%	6.81%
T = 0 h, M = 20%, W = 18%	0.88%	1.34%	2.62%	5.37%	6.03%	10.62%	13.63%
T = 12 h, M = 10%, W = 18%	—	—	—	—	—	—	0.11%
T = 12 h, M = 15%, W = 18%	—	—	—	—	—	—	0.21%
T = 12 h, M = 20%, W = 18%	—	—	—	—	—	—	0.39%
T = 24 h, M = 10%, W = 18%	—	—	—	—	—	—	0.08%
T = 24 h, M = 15%, W = 18%	—	—	—	—	—	—	0.17%
T = 24 h, M = 20%, W = 18%	—	—	—	—	—	—	0.31%
T = 48 h, M = 10%, W = 18%	—	—	—	—	—	—	0.07%
T = 48 h, M = 15%, W = 18%	—	—	—	—	—	—	0.16%
T = 48 h, M = 20%, W = 18%	—	—	—	—	—	—	0.29%
T = 72 h, M = 10%, W = 18%	—	—	—	—	—	—	0.07%
T = 72 h, M = 15%, W = 18%	—	—	—	—	—	—	0.15%
T = 72 h, M = 20%, W = 18%	—	—	—	—	—	—	0.24%
T = 0 h, M = 20%, W = 14%	3.24%	8.93%	11.85%	—	—	—	—
T = 0 h, M = 20%, W = 22%	0.45%	1.00%	2.26%	3.23%	3.58%	4.27%	6.05%

## Data Availability

The data provided in this study can be obtained from the second author Huicong Su.

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
