# Peer review of "Experimental Study on the Cracking and Mechanical Properties of Lime Soil with Different Slaking Conditions of Newly Repaired Earthen City Walls"

_materials, 2022, doi:10.3390/ma15124151_

Round 1

Reviewer 1 Report

Very interesting paper, I really  enjoy. However, the following minor correction has been made before publication.

The introduction is too short, the authors only reference a few paste studies. It is recommended to add updated references on similar topics

At the end of the introduction indicated which is differences between this and other similar articles. Better to add research significance.

Table 2 why composition is less than 100%.

Formatting issue (Error source in shear strength section).

Compared the results obtained by past studies.

Please indicate the drawbacks/limitations of this study.

Highlights the future aspects.

Author Response

Dear Reviewer 1:

  Thanks for your comments on our paper. We have revised our paper according to yours comments.

  We have checked the manuscript and revised it according to the comments. We submit here the revised manuscript. Please see the attachment.

  If you have any question about this paper, please don’t hesitate to let me know.

Sincerely yours

Dr. Xu

Reviewer 2 Report

Paper “Experimental study of the cracking and mechanical properties of lime soil with different slacking conditions of newly repaired earthen city walls” researches crack resistance, hardness of earthen walls in dependence on lime content, slacking time, and moisture. The authors have determined important parameters that will help to restructure earthen walls and keep important archeological and cultural buildings and other objects. I recommend only minor revision.

  1. Authors write that the clay composition was determined by X-ray. Usually the exact model is required in the form “.. by the X-ray EQUIPEMENT MODEL (MANUFACTURER, COUNTRY, CITY)”. This concerns all mentioned methods – for each authors should indicate model, manufacturer, country and city of all used equipment. If you describe exact equipment later, you should indicate it a first mention of the method (for example, X-ray analysis (see paragraph 2.2.7)”). For example, “constant humidity and temperature box” – what kind of box & who produced it? In what rates the box can keep temperature and humidity constant? All this is quite important. Also I would like to know for tables 1 and 2 – how many samples, from which exactly places were taken. What is the error for determination of average between all these samples? If the sample is only one, please explain why.
  2. The paragraphs 2.2.2. and 3.4.2. contains “Error! reference source not found” instead of a reference.
  3. Fig. 10a contains red circles outside the figure. Or the circles have run away from the figure, or the figure was initially bigger than it is now.
  4. I recommend adding DOI where possible to all cited papers.
  5. I have mentioned several insufficient text mistakes – missing gap, extra gap, extra comma etc. Please look through your paper attentively and eliminate these mistakes.

Author Response

Dear Reviewer 2:

  Thanks for your comments on our paper. We have revised our paper according to yours comments.

  We have checked the manuscript and revised it according to the comments. We submit here the revised manuscript. Please see the attachment.

  If you have any question about this paper, please don’t hesitate to let me know.

Sincerely yours

Dr. Xu

Reviewer 3 Report

Manuscript Title: “Experimental study on the cracking and mechanical properties of lime soil with different slaking conditions of newly repaired earthen city walls”

  1. The entire manuscript is not following MDPI, 2022 format.
  2. The problem statement is not clear in the abstract.
  3. In the abstract, it is recommended to state the designed slaking times.
  4. In the abstract, remove and after cohesion. The words “are decrease” in the same line should be are decreased. Check the grammar and English language throughout the manuscript.
  5. It is recommended to split the results presented in the abstract into two or three sentences.
  6. It is recommended to add slaking condition into the keywords.
  7. Introduction must be improved by stating the state-of-art and clearly coming up with the gap in the research.
  8. Results and discussion are clear, however, it is recommended to merge the results and discussion.
  9. Correct the “Error! Reference source not found “that appeared on page 11.
  10. Check Fig. 10, some circles outside the picture.
  11. It is recommended to list the last point (Proper slaking … “in the conclusion as point (4).

Consequently, the research area is worthy of investigation. Therefore, the reviewer is recommending the manuscript after minor revision.

Author Response

Dear Reviewer 3:

  Thanks for your comments on our paper. We have revised our paper according to yours comments.

  We have checked the manuscript and revised it according to the comments. We submit here the revised manuscript. Please see the attachment.

  If you have any question about this paper, please don’t hesitate to let me know.

Sincerely yours

Dr. Xu

Reviewer 4 Report

This paper extensively compiled and argued the results with comparing the influence of slaking condition on the cracking and mechanical properties of lime soil for the repaired Kaifeng city wall, specimens with different limes, waters and slaking times and characterization studies were analyzed. These are my observations,

Ø  The introduction part is not explicitly written up to the scientific required.

Ø  There is need of strong literature support with justification of novelty of work.

Ø  Materials basic properties are not clearly discussed.

Ø  Add some more literature in the introduction

Ø  Add Physical characterization of lime soil

Ø  If possible, add a sequence of research methods.

Ø  In keywords using “Carbonization reaction.”  As you discussed in this paper?

Ø  Citations are not sequence and clear.

Author Response

Dear Reviewer 4:

  Thanks for your comments on our paper. We have revised our paper according to yours comments.

  We have checked the manuscript and revised it according to the comments. We submit here the revised manuscript. Please see the attachment.

  If you have any question about this paper, please don’t hesitate to let me know.

Sincerely yours

Dr. Xu

Round 2

Reviewer 4 Report

This paper extensively compiled and argued the results were analysed,

Ø  The introduction part is not explicitly written up to the scientific required.

Ø  There is need of strong literature support with justification of novelty of work.

Ø  Materials basic properties are not clearly discussed.

Ø  Add some more literature in the introduction

Author Response

(The authors gave the same response as above.)
